# What Are the Recipes of an Entrepreneur’s Subjective Well-Being? A Fuzzy-Set Approach for China

**DOI:** 10.3390/ijerph20010417

**Published:** 2022-12-27

**Authors:** Zihan Yang, Xu Cai, Yujia Jiang, Guobiao Li, Guojing Zhao, Peng Wang, Zhaoxin Huang

**Affiliations:** 1College of Innovation and Entrepreneurship Education, Wenzhou Medical University, Wenzhou 325035, China; 2College of Politics and Public Administration, Shandong Youth University of Political Science, Jinan 250103, China

**Keywords:** entrepreneurs, subjective well-being, physical and mental health, social capital, FsQCA

## Abstract

Entrepreneurs face more pressure and challenges than ordinary workers, which has a serious impact on their physical and mental health. Therefore, the research focus has gradually shifted from objective indicators of entrepreneurial performance to exploration of entrepreneurs’ subjective well-being. However, previous studies were often limited to a net effect analysis of a single dimension under symmetric thinking in quantitative research. Therefore, this study uses fuzzy-set qualitative comparative analysis (fsQCA) to analyze the configuration path of entrepreneurs’ physical and mental health at the individual level, social capital at the collective level, and subjective well-being from the perspective of configuration. The sample was of 279 effective entrepreneurs from the 2017 China General Social Survey (CGSS). Four types of entrepreneurs were found to improve their high well-being profiles: optimistic efficiency-driven, trust efficiency-driven, strong psychology-driven, and weak relationship-driven. Research shows that the interaction between physical and mental health and social capital jointly affects the subjective well-being of entrepreneurs. The research findings reinforce the need for attention to the physical and mental health of entrepreneurs, which are conducive to their active participation in social life. Additionally, establishing weak relationship-oriented interpersonal networks and accumulating social resources to further achieve higher subjective well-being is required.

## 1. Introduction

According to the Global Entrepreneurship Monitor report (GEM), the proportion of entrepreneurs in all countries is increasing, as is the rate of entrepreneurship [1]. Entrepreneurial activities inject new vitality and development into the social economy; however, entrepreneurs have become a “three high” group with high risks, high intensity, and high challenges. Their physical and mental health status is generally higher than that of professional workers, which has attracted the attention of researchers to their physical and mental health. Well-being, as the main index to judge the healthy life status of individuals, has become a key dependent variable for scholars studying entrepreneurs [2]. Since 2009, the rapid growth of articles on topics such as “entrepreneurs” and “happiness” in high-impact journals has been sufficient to demonstrate the importance of this research field [3].

Existing studies have analyzed the factors influencing entrepreneurs’ subjective well-being at the individual and collective levels. On the individual level, demographic variables such as gender, education, marital status, number of children, and personal income are significantly correlated with the happiness of entrepreneurs [4]. Mahadea and Ramroop [5] selected entrepreneurs from the business council through random sampling and selected employees from different enterprises for horizontal comparative research. They found that gender, personal education status, marriage, and having children are significantly related to the happiness of employees and entrepreneurs. At the same time, the results of the variance analysis show that the overall happiness level of entrepreneurs is higher than that of employees because of their greater independent decision-making power. Entrepreneurs, who are educated, married and have one–three children report higher levels of subjective well-being (SWB). Further, psychological capital, such as self-efficacy, optimism, and hope, is significantly correlated to the happiness of entrepreneurs [6,7]. Baron et al. found that entrepreneurs’ strong psychological capital improved their ability to withstand and manage pressure, thus reducing their perception of pressure and significantly enhancing their SWB [8]. At the collective level, there is a correlation between the internal organizational environment, external geographic environment, and happiness of entrepreneurs. Crum and Chen [9] found that good entrepreneurial institution quality and systems have a significant positive impact on entrepreneurs’ happiness. Based on longitudinal data from the UK Household Longitudinal Study (UKHLS), Abreu et al. [10] found that the happiness level of entrepreneurs is subject to their geographical location; the better the external entrepreneurial environment, the happier entrepreneurs will be. Additionally, the living environment, based on institutional trust, social trust, and community cultural atmosphere, is significantly correlated with the happiness of entrepreneurs [11,12,13]. The research found that institutional trust in government agencies, social trust in interpersonal relationships, and family support significantly predicted job satisfaction and life satisfaction of entrepreneurs, and positively improved their SWB [2].

Existing studies have explored the impact of entrepreneurs’ personal characteristics and internal and external entrepreneurial environments on happiness and made certain contributions to the countermeasures as well as suggestions on how to improve entrepreneurs’ happiness. However, most studies have focused on quantitative methods based on the linear relationship between variables of a single dimension, ignoring the interaction and influence of factors affecting happiness at all levels.

After World War II, research on human happiness emphasized that it was related to physical diseases; however, the emergence of positive psychology elevated the research from the consideration of physical health factors to the exploration of psychological health factors. Seligman et al. found that the key to disease prevention is to cultivate people’s positive qualities in addition to general drug means. Such positive qualities include hope, optimism, courage, and faith [14]. Positive psychology is a kind of subjective positive experience of people at present, which is the satisfaction of the past and present life, as well as a positive and optimistic vision and hope for the future [15]. SWB, optimism, happiness, and self-determination are four elements of positive psychology. They interact with each other to promote people’s happiness and well-being and have a direct positive impact on physical health [16,17]. Positive psychology is a revolution in the field of psychology that advocates people’s positive lifestyles and happiness experiences. Thus, this study examines the relationship between entrepreneurs’ physical and mental health, and SWB.

Simultaneously, people have begun to pay attention to the influence of social capital following the development of the social economy and the gradual formation of social networks; notably, physical and mental health and social capital are closely related [18,19,20]. Therefore, research on entrepreneurial well-being should not be limited to a single linear regression and should consider the causal relationship between variables as well as the interaction between different configurations, comprehensively examining the impact of each variable on well-being.

In this study, based on the theory of configuration related to positive psychology and social capital theory and using the fuzzy integrating qualitative comparison analysis (fsQCA), we examine the relationship between physical and mental health and social capital and entrepreneurs’ complex causal relationship with well-being. Furthermore, we explore how entrepreneurs’ physical and mental health interact with social capital to enhance their happiness.

The study makes the following contributions to literature: regarding research methods, fsQCA is used to examine the sufficient and necessary conditions to improve entrepreneurial well-being; the configuration matching mode is used to expand the method in this field. Regarding the research content, the two factors affecting well-being, physical and mental health, and social capital, are integrated. Based on positive psychology and social capital, the influence of the interaction between individual internal characteristics and external environmental resources on well-being is explored, and the research on the combination model of conditional variables is broadened. Following the latest research trends, this study focuses on entrepreneurs. This study helps to alleviate entrepreneurial anxiety, reduce the rate and prevalence of poor health among entrepreneurs, promote the effective development of entrepreneurial activities, and further promote social and economic well-being through the needs of the times and academic field research tendency.

## 2. Literature Review and Theoretical Framework

### 2.1. SWB of Entrepreneur

Research on well-being is mostly based on the subjective self-assessment of the research objects. This study traces the previous literature, defines entrepreneurial well-being using the concept of subjective well-being and further evaluates it. Chekola defines SWB as the satisfaction of one’s desire and goal [21]; Bradburn emphasizes positive emotions and pleasant emotional experience [22]; and Shin and Johnson indicate that it is a comprehensive assessment of one’s quality of life according to one’s choice criteria [23]. This study defines SWB based on Diener [24], stating that it is an emotional and cognitive evaluation of one’s life quality in a long-term stable period, a subjective, overall, and stable concept.

Researchers have explored the influencing factors of entrepreneurial well-being from different perspectives, which can be roughly divided into economic and non-economic factors. Among non-economic factors, initially, based on individual characteristics, the author examines the influence of gender, age, religion, education, marital status, as well as level of physical health on subjective entrepreneurial well-being [8]. Then, mainly by focusing on the correlation between macro institutional environment and well-being, in a formal institutional environment, the quality of entrepreneur-friendly institutions and size of business environment index are significantly positively correlated with entrepreneurial well-being [25]; in an informal institutional environment, the social legitimacy of entrepreneurship and entrepreneurial culture influence people’s attitudes and evaluation of entrepreneurial activities, which consequently affects the perception of entrepreneurial well-being, and has a positive or negative impact on entrepreneurial behavior [13,26]. Among economic factors, an increase in entrepreneurs’ income improves their SWB significantly [27]. Meanwhile, entrepreneurial performance evaluated based on objective financial indicators has a positive impact on well-being [28].

Based on previous studies, this study focuses on non-economic factors as the entry point, not limited to the single dimension of variables and quantitative linear thinking research mode, and further examines the interaction between individual physical and mental health and collective social capital on entrepreneurial SWB.

### 2.2. The Physical and Mental Health and SWB of Entrepreneurs

The World Health Organization defines health as not only the absence of disease but also a state of complete physical, psychological, and social adaptation [29]. Physical and mental health are the main influencing factors of SWB [19]. Therefore, exploring the relationship between physical and mental health and well-being has practical significance.

Related studies have found a positive correlation between physical health and well-being indicators [30]. Poor health has a significant negative impact on well-being [31]. Moreover, self-rated health level, which focuses on the body, is the determinant of SWB [32]. However, scholars indicate that self-entrepreneurship may affect the physical health of entrepreneurs [33]. Compared with non-entrepreneurs, the physical condition of entrepreneurs is worse, affecting their well-being [34]. Therefore, we should consider the relationship between the physical health of entrepreneurs and their SWB and attempt to eliminate the adverse effects of poor physical health on their well-being and entrepreneurial activities.

Mental health is the continuous and positive development of a psychological state. Moreover, individuals with correct self-recognition, reasonable will, and positive attitude can promote the emergence of reasonable emotion and correct behavior. Related research shows that entrepreneurs’ strong psychological capital, together with self-efficacy, optimism, hope, and resilience will enhance their tolerance to stress, resulting in low perceived stress levels and significantly improved SWB, under strong psychological capital [8]. A healthy psychological state not only improves entrepreneurial well-being but also promotes the enhancement of entrepreneurial intention and performance, which further plays an important role in entrepreneurial outcomes [35,36,37]. However, previous studies indicate that entrepreneurs are more likely to have two strong emotional fluctuations, positive and negative, than non-entrepreneurs, and that such fluctuations are significantly negatively correlated with well-being [38]. To avoid excessive negative emotions, it is necessary to pay attention to the continuous and stable positive psychological state of entrepreneurs, which is relatively important for them to conduct entrepreneurial activities effectively.

On the one hand, self-efficacy refers to people’s confidence in their ability to use their skills to conduct and complete a certain activity [39], which affects people’s overall thinking, emotions, and psychosomatic reactions [40]. In the entrepreneurial process of constantly pursuing goal setting and self-decision-making, entrepreneurs stimulate self-determination, generating a higher sense of continuous self-efficacy than other groups [6]. Such a stable level of self-efficacy promotes job satisfaction and positive SWB, and further influences entrepreneurial behavior [7,41,42].

On the other hand, optimism is a positive personality factor, that can be maintained easily during successful periods. Adversity helps to maintain a positive spirit and good psychological state of courage. It is generally believed that most entrepreneurial behavioral characteristics are driven by optimism and confidence [43,44]. Furthermore, many related studies have found a significant positive correlation between optimism and SWB [45]. The more optimistic individuals are, the higher their SWB level is [46,47]. Therefore, it is necessary to explore whether optimism has a lasting effect on well-being, and whether it promotes the smooth development of entrepreneurial activities during the entrepreneurial process.

In summary, self-efficacy and optimism are the main manifestations of mental health and emotions. Based on positive psychology, if people can endure adversity and pressure calmly, and possess positive qualities such as self-confidence and optimism, they will evoke their positive emotions and induce well-being [14]. This study selected the above two dimensions as the evaluation factors of mental health to explore their impact on the SWB of entrepreneurs.

### 2.3. Social Capital and SWB of Entrepreneurs

Regarding the definition of social capital, Putnam indicates that social capital refers to the characteristics of social organizations, namely networks, norms, and social trust, which promote coordination and cooperation, and achieve mutual benefit through collective action [48]. Nahapiet and Ghoshal defined social capital in three dimensions: structural, relational, and cognitive [49]. The structural dimension focuses on the connection between individuals and units, while the relational dimension mainly illustrates the personal relationships built through mutual interaction. The cognitive dimension is reflected in the common vision and values formed by individuals and collectives [50]. Social capital theory holds that the contact and interaction between individuals establish close social relations, and further build corresponding social networks; they obtain the resources owned by the group, support and benefit each other, and jointly realize economic progress and social development [51]. Social trust and community participation promote the formation of social networks and increase individual social capital, which has been proven to be ultimately significantly correlated with SWB [52,53]. Similarly, social capital is divided into cognitive and structural social capital based on the definitions of various scholars [54], and we explore the correlation between social capital and the SWB of entrepreneurs.

#### 2.3.1. Cognitive Social Capital

Cognitive social capital refers to feelings and values shared between subjects, such as trust [55]. People promote exchange and cooperation based on trust, form cohesion and social power to solve conflicts, and achieve mutual interests [56]. Research on social capital and well-being in the US shows that there is a strong correlation between social trust and well-being, and that the two have a strong positive externality effect [57]. Research in South Korea also showed that cognitive social capital is significantly positively correlated with well-being [32]. When exploring the influencing factors of entrepreneurial SWB in China, Xu et al. found that a macro-entrepreneurial environment based on institutional trust significantly predicted the well-being of entrepreneurs [2]. Further, the existing literature confirms that the reduction of social capital with social trust as the main content reduces SWB [58]. The influence of cognitive social capital on SWB cannot be ignored, especially for entrepreneurs, as trust can effectively encourage them to cooperate despite risks and challenges to achieve a sense of achievement and harvest well-being in a harmonious entrepreneurial atmosphere.

#### 2.3.2. Structural Social Capital

Structural social capital refers to objectively formed social organizations and social networks in which people play different roles, establish connections between people or between people and groups, and achieve collective action goals [55]. Agampod et al. divided membership of social organizations and frequency of social participation into two dimensions [59]. Related studies showed that cohesion and interaction among social members have a positive impact on SWB [60]; the perception of the social support of residents and dependence on community and benefit-sharing among residents also play a role in the residents’ SWB and affect their well-being index [61]. Research on the relationship between economic and social capital and SWB in Europe shows that social and institutional trust has a significant positive impact on SWB. However, the most significant influencing factor of SWB is the strong intimate social relationship based on social interaction and social integration, followed by economic factors [62]. Among family, neighbors, and friends, informal social interaction often has a significant positive influence on SWB. However, formal organizational participation is significantly negatively correlated with individual well-being [57], highlighting the importance of informal relationship networks in accumulating personal social capital and enhancing well-being.

Based on previous literature, this study selects physical and mental health and social capital to explore the influencing factors of the SWB, examines the causal relationship between the two dimensions and SWB, and explores the adequacy or necessity conditions and configuration effects that affect SWB from the perspective of configuration. The research model is illustrated in Figure 1.

## 3. Research Methods

### 3.1. The Data Source

This study uses data from the Chinese General Social Survey (CGSS) in 2017. The CGSS is the earliest national, comprehensive, and academic survey project in China. The CGSS collects comprehensive and diverse data on society, family, and individuals, providing sufficient and effective data sources for researchers to explore various social problems arising in the process of social change, as well as data for international comparative studies. Thus far, it has included national data from 2003 to 2017. This study uses the latest CGSS data from 2017, and further selects 279 effective entrepreneurs (62% male; 38% female) for fsQCA data analysis. These entrepreneurs are located in 28 provinces or cities in China, such as Beijing, Shanghai, and Zhejiang, with a relatively large proportion in Beijing, Hunan, and Sichuan Province. Therefore, the sample has a strong national representative of entrepreneurs. Furthermore, it is helpful to explore the factors influencing entrepreneurs’ SWB in the context of the Chinese culture.

### 3.2. Measured Variables

The study selected the CGSS questionnaire item, “Which of the following scenarios is more consistent with your current work situation?” which had two options, “I am a boss (or a partner)” and “Individual industrial and commercial households” during screening. The sample data of 290 entrepreneurs were further obtained. As for the measurement of SWB, this study selected the CGSS data, “generally speaking, do you think your life is happy”, as a problem term. However, is it reliable to measure happiness using a single question? This study provides an answer from the following literature. Robinson et al. [63] conducted a systematic study on reliability problems. They believed that the above measurement items were reliable for stable and repeatable experiments. Wilson et al. [64] compared expert assessment results with interviewees’ self-assessment results and found that the two were very similar; the validity of this measurement has been confirmed. On this basis, this method of measuring SWB with single questions has become the most widely used method in the world [65], and researchers using CGSS data chose this single item to measure SWB [66,67].

In terms of the measurement of social capital, this study draws on the definition of Krishna and Uphoff [54] and divides social capital into cognitive social capital and structural social capital. Social trust and social networks are used as the measurement criteria. Social networks are embodied in people obtaining resources and achieving goals through interactions with relatives and neighbors. The CGSS (2017) data contain measurement items that conform to this definition. This study extends the selection of measurement items by previous researchers for social trust and social networks in the application of CGSS data [68,69,70,71]. In terms of self-efficacy measurement, the CGSS (2017) data refer to the mature self-efficacy scale designed by Zhang and Chen. The measurement dimensions focus on three levels: achieving goals, overcoming difficulties, and solving problems. The scale has good reliability and validity. Therefore, the self-efficacy measurement items in the 2017 CGSS data can be used in existing studies [72,73]. Based on the measure of optimism, CGSS (2017) refers to the Life Orientation Test (LOT) and its revised version known as Revised Life Orientation Test (LOT-R) developed by Scheier and Carver based on the direct belief model, which measures the “individual’s expectation of the outcome of an event (positive and negative attitudes),” the most widely used scale at present. It has good reliability and validity and is applicable to existing studies [74,75]. Table 1 lists the variables and their corresponding items.

The SWB measurement item was assigned scores of 1–5; the higher the score, the stronger the sense of well-being. Physical and mental health is divided into physical health and mental health and classified into two levels, physical health (PH) measurement item (scores 1–5), with higher scores indicating better physical health; mental health includes self-efficacy (SE) and optimism (OM). Self-efficacy had scores 1–8; the higher the score, the stronger the self-efficacy. Six items measuring optimism were assigned scores of 1–5. The lower the score of the first three items, the more optimistic they were, the higher the score of the last three items, the more optimistic they were. Social capital includes both cognitive and structural social capital. Cognitive social capital evaluates social trust (ST) and had scores of 1–5. The higher the score, the higher the degree of social trust. Structured social capital includes neighbor interaction (NI) and friend interaction (FI). The measurement items are rated from 1 to 7, with higher scores indicating more frequent interactions. All the samples with incomplete variable data were eliminated; 279 samples of effective entrepreneurs were ultimately obtained.

### 3.3. fsQCA

Traditional quantitative research methods, based on reductionist assumptions, only analyze the simple symmetric linear relationship between individual antecedents and results [76]. However, complex social phenomena reflect a complex relationship between multiple concurrent conditions and results. It is necessary to clarify the causal logic to explain these complex management phenomena scientifically. Qualitative comparative analysis of fuzzy sets (fsQCA) is a method used to establish a causality model based on set theory and Boolean operation [77]. It is case-oriented, based on configuration theory, expressed by set relations, and studies the complex causality between variables (necessary, and sufficient conditions). Compared with traditional symmetric quantitative methods, fsQCA has the advantage of avoiding the endogenous problems of reverse causality, missing variable bias, and sample selection bias from the source [78]. It allows for the exploration of the complex impact of different configuration combinations formed by antecedent conditions in the case of interdependence and interaction on result variables, mining of the core and edge conditions affecting the result variables, and finding the complexity and diversity of complementarity, substitution, or inhibition between different configurations, from the perspective of configuration [79]. fsQCA is not only suitable for small case studies with 10–15 samples but is also suitable for medium samples with 15–50 samples and large samples with more than 100 samples. It integrates the dual advantages of quantitative and qualitative methods to solve different causal problems [79,80]. Based on the fsQCA method, this study explores the factors influencing entrepreneurial SWB from the perspective of configuration. Based on the positive psychology and social capital theories, the complex influence of the physical and mental health and social capital on SWB of entrepreneurs is examined using 279 entrepreneurial samples from CGSS data.

## 4. Analysis

### 4.1. The Calibration Data

The first step to apply the fsQCA method is fuzzy set calibration, that is, regular values are transformed into fuzzy variables with values between 0 and 1; three critical values are set: full membership, crossover membership, and full non-membership. Previous studies show that for large sample variables, upper and lower quartiles, upper or lower deciles, and 5th and 95th percentiles are generally used as anchors for calibration [77,79,81]. In this study, the 5th, 50th, and 95th percentile were used as anchor points to calibrate conditional and result variables. Table 2 shows the calibration values.

### 4.2. Necessity Analysis

Necessity and adequacy analyses are the basis for configuration path analysis [81]. Before adequacy analysis, the fsQCA software was first used to conduct necessity analysis on variable data, to further test whether antecedent conditions and their negation are significant influencing conditions of the result variables. In fsQCA operation methods, consistency values are typically used to evaluate the degree of recognition of the subset relations [82]. For example, the antecedent conditions (X) in this study include physical health, self-efficacy, optimism, social trust, NI, and friend interaction, where the outcome condition (Y) is SWB. We must determine whether these antecedent conditions X constitute a set subset of Y and the extent to which they affect Y. In this set relation, the cases have different degrees of membership, and the membership score ranges from 0.0 to 1.0. The closer the score is to 1, the more X belongs to the outcome condition Y. We use 0.9 as the dividing standard. If the consistency score exceeds the threshold value of 0.9, this condition is called “necessary”; that is, X is a necessary condition for Y [83]. Therefore, we conducted a necessity analysis of high SWB and non-high SWB. Table 3 shows the results of the necessity analysis. As seen in the table, the consistency value of the influence of a single antecedent condition on SWB was not more than 0.9, indicating that the explanatory power of every single condition on the SWB of entrepreneurs was weak, which was not enough to constitute the necessary condition of the outcome variable.

### 4.3. Adequacy Analysis

The fsQCA software was used to construct the truth table for the data adequacy analysis. Following the research of Fiss and other scholars [79], the consistency and PRI thresholds were set as 0.8, and 0.7, respectively, to achieve the purpose of reducing contradictory configurations. There are three types of solutions: complex, parsimonious, and intermediate solutions. Based on the research of Ragin [77], the intermediate solution is regarded as the main result of data analysis, and the conditional variables that appear in both the parsimonious and intermediate solutions are regarded as the core condition, and the conditional variables that only appear in the intermediate solution are regarded as the edge condition. Based on this, four configuration paths affecting the high SWB of entrepreneurs 2343 obtained, as shown in Table 4.

The results show that the overall consistency of the model is 0.923, indicating that the four interpretation paths cover approximately 92% of the cases, which is a sufficient condition for causal reasoning. The coverage of the model is 0.531, indicating that it explains approximately 53% of the factors affecting the SWB of entrepreneurs. Further, the consistency of the four configurations in the model was 0.925, 0.974, 0.971, and 0.946, respectively, which were all approximately 0.9, indicating that there was a good subset relationship with the SWB of entrepreneurs; in other words, the antecedent conditions had good explanatory power for the outcome variables.

### 4.4. Robustness Test

We tested the robustness of the antecedent configuration of a high SWB. First, a configuration analysis was conducted with non-high SWB as the outcome variable. The results showed that the configuration leading to high SWB was completely different from that of non-high SWB, and that the condition combination was not symmetrical, relatively reflecting the robustness of the results [84] (Table A1). Second, the case consistency threshold increased from 0.8 to 0.85, resulting in completely consistent configurations (Table A2). Finally, 279 cases were randomly deleted, 25 random cases were deleted, and the configurations obtained from the analysis were consistent [85,86] (Table A3). The robustness tests yielded robust results.

## 5. Results

Based on CGSS effective entrepreneurs’ data, we used the fsQCA software to analyze entrepreneurs’ physical and mental health and social capital combination of SWB, and concluded four pathways for entrepreneurs to generate high SWB, respectively, to explain the path for the corresponding interpretation.

### 5.1. Optimistic Efficiency-Driven Entrepreneurs

Configuration 1 was PH × OM × SE ×~ NI ×~ FI, with a consistency of 0.925, covering 92.5% of the cases. The coverage was 0.398, which explained 39.8% of entrepreneurs’ high SWB. Configuration 1 shows that entrepreneurs will have higher SWB when they have good physical health and hold a positively optimistic attitude and strong self-identity. It can induce strong well-being, despite the entrepreneur having too little interaction with his neighbors and friends as long as they maintain good health, have the right subjective judgment of their ability, and have a positive mood. According to positive psychology theory [14], when people have more positive emotions such as pleasure, interest, pride, and satisfaction, they block their negative emotions such as sorrow, sadness, and anxiety, and increase their positive subjective experience of life and satisfaction, resulting in well-being. Diener’s research focuses on the influencing factors of SWB, posits that in addition to individual physical quality and personality traits, the macro social environment also has an impact on people’s SWB. However, the results found that the key to determining people’s SWB is not the occurrence of external events, but people’s views on the events. In other words, it comes from an individual’s subjective cognition and feelings regarding their own life and events [87]. Therefore, optimism and self-efficacy are key influencing factors of SWB. Further, previous related studies show that both physical and mental health are positively correlated with individual SWB [88,89], and that good mental health is usually associated with a good physical state [90], which confirms this study’s research results; physical and mental health of entrepreneurs interact and jointly affect their SWB, rather than just exhibiting a single linear correlation.

### 5.2. Trust Efficiency-Driven Entrepreneurs

Configuration 2: PH × SE × ST ×~ NI ×~ FI, the consistency of the configuration was 0.974, covering 97.4% of the cases. The coverage is 0.356, which explains the high SWB of 35.6% of entrepreneurs. Configuration 2 showed that entrepreneurs with good physical fitness, a strong sense of self-efficacy, and a sufficiently high level of trust in society would have a higher SWB. Compared with Configuration 1, Configuration 2 highlighted the key effects of self-efficacy and social trust on SWB. Entrepreneurs often stimulate their firm will, confidence, and determination when faced with high-risk, challenging entrepreneurial activities, and an unstable entrepreneurial environment. They have a higher sense of self-efficacy than ordinary people and believe that they are capable of completing tasks and being successful, to satisfy their needs of autonomy, competence, and belonging; hence enhancing their SWB [7]. Meanwhile, previous studies show that social trust has a significant positive impact on SWB [91,92]. Individuals with a high degree of social trust have a strong sense belonging in the community and surrounding environment, which helps generate a high level of social cohesion between groups, to further establish good interpersonal relationships while accepting and including others and connect groups and effective resources. The whole environmental climate and development process improve living standards and enhance life satisfaction and have a fundamental impact on people’s SWB. Therefore, even though entrepreneurs do not interact with neighbors and friends frequently, their strong sense of self-efficacy and high level of trust in society will promote their positive emotions and good interpersonal relationship atmosphere perception, thus influencing their SWB.

### 5.3. Strong Psychology-Driven Entrepreneurs

Configuration 3: PH × OM × SE × ST ×~ NI, the consistency of the configuration was 0.971, covering 97.1% of the cases. The coverage was 0.341, which explains the high SWB of 34.1% of entrepreneurs. The results showed that entrepreneurs with good physical and mental health and adequate cognitive social capital have higher SWB. The core conditions of configurations 2 and 3 were the same, that is, good physical state, high self-efficacy, and social trust. However, the difference was that in configuration 3, optimism and non-high neighbor interaction existed as marginal conditions. Previous studies show that optimism makes people more prone to trust, and more optimistic people are more likely to trust others [57,93]. Similarly, social trust is closely related to physical and mental health. Social trust can relatively reduce the prevalence of diseases and promote physical health as well as reduce people’s negative emotions such as anxiety [94], and improve mental health [95,96]. The higher the level of social trust, the stronger the individual sense of security. This plays a positive role in expanding individual social networks and improving life quality, and further improves SWB. Relying on trust to obtain required resources and establishing harmonious interpersonal relationships is conducive to the effective development of entrepreneurial activities and improves life satisfaction and well-being, especially for entrepreneurs. The results show that entrepreneurs with a strong psychological state with a high level of mental health and social trust will have better well-being. This corresponds to the conclusions of previous studies, and based on the previous single linear variable analysis, summarizes the interdependent and mutually affecting symbiotic relationships among physical health, mental health, and social trust in the form of conditional combination, which play a role in promoting entrepreneurs’ high SWB.

### 5.4. Weak Relationship-Driven Entrepreneurs

Configuration 4 was PH × OM × SE ×~ ST × NI × FI, with a consistency of 0.946, covering 94.6% of the cases. The coverage was 0.320, which explains approximately 32% of entrepreneurs’ high SWB. The results showed that entrepreneurs with good physical and mental health and adequate structural social capital would have higher SWB. The physical and mental health level of physical health, optimism, self-efficacy, and structural social capital of neighbor interaction and friend interaction are the core conditions. When entrepreneurs have a strong physique, an optimistic state of mind, and charming self-confidence, and are happy to get along with neighbors and friends frequently in daily life, they can obtain high SWB even if they do not have enough social trust. Based on social capital theory, individuals are mosaics in the structure of the social network formed by the interaction between the person and the group, and individuals can acquire corresponding social resources and accumulate their rights, wealth, and prestige under certain rules, to meet the needs of their development, improve their socioeconomic status, and enhance job satisfaction, life satisfaction, and SWB [51]. Further, previous related studies show that social resources generated by interaction with neighbors and friends will improve people’s self-esteem and social cohesion, and indirectly positively impact their life satisfaction and health [97,98]. Corresponding to configuration 4, although entrepreneurial social trust is insufficient, the process of long-term interaction with neighbors and friends increases the perception of social connection and safety under weak relationships, which is beneficial to the development of physical and mental health, and indirectly promotes the production of cognitive social capital, which further impacts entrepreneurial SWB. Good social relationships and frequent community participation interact with physical and mental health and jointly promote high entrepreneurial SWB.

According to the data analysis results in this study, physical health and self-efficacy both appeared as core conditions in each interpretation path, while optimism appeared as the core and marginal conditions in the three configurations, respectively. Evidently, physical and mental health is key to promote entrepreneurs’ SWB. Good health, an optimistic attitude, and the quality of being assertive are often the main influencing factors of well-being. Second, social trust as the core condition appeared in the two profiles: high cognitive social capital can better establish trust between people; the increase of belonging and security promotes the cooperation between entrepreneurs and other people, and resource sharing still positively affects the SWB of entrepreneurs. Finally, neighbor and friend interactions emerged as the core conditions in configuration 4. Sufficient structural social capital relatively augments entrepreneurial social resources, which can be financially or emotionally supported and assisted by the informal relationship network, and similarly contribute to high entrepreneurial SWB.

## 6. Discussion

### 6.1. Theoretical Contributions

First, the research results of Europe and the US emphasize the decisive role of economic income level on well-being [99,100]. Based on the theory of positive psychology, this study proves that physical and mental health are the core conditions that affect the entrepreneurial SWB. In the Chinese context, the SWB of Chinese people is significantly influenced by traditional Chinese philosophy. SWB in Confucianism, Taoism, and Buddhism emphasize collective well-being and social harmony based on collectivism [101]. The well-being of Chinese people is weakly correlated with material enjoyment, but emphasizes spiritual fulfillment and satisfaction, with belief and emotion as key elements [102]. Studies on Chinese residents confirmed that spiritual enrichment, satisfaction, and long-term positive emotions are the key influencing factors of well-being [61].

Second, from the dimension of social capital, previous related studies show that both social trust and social networks significantly improve individual SWB [62,103]. Interestingly, among the four configuration paths in this study, the first three show that the social network based on neighbor and friend interaction is a non-core condition or a non-marginal condition that affects SWB, indicating that entrepreneurs have no significant impact on their SWB, despite not interacting with neighbors and friends. This is a deviation from previous research conclusions. However, this study still provides corresponding explanations based on the Chinese context. Tracing back to traditional Chinese society, residents tend to live together with their families, forming a lifestyle and cultural pattern with the family and clan as the main body, resulting in a high sense of identification with blood ties and geographical relations [104]. Evidently, Chinese people are more inclined to regard blood relationships, geographical relationships, and other similar relationships as communication principles to establish trust and intimate social relationship networks and generate collective well-being [105]. However, it is worth noting that the last configuration path shows that both neighbor and friend interactions are the core conditions for generating high SWB, with an explanation degree of 32%. This path is consistent with previous research results, and based on the social capital theory and the development trend of modern society, people’s contact with colleagues, neighbors, and friends from all over the country; in the interaction and exchange of information, such as better interaction and reciprocity of collective behavior, construct informal relationships other than familial relationships. This network environment gives people a sense of belonging, security, positive emotions, and non-kinship spiritual satisfaction, and improves SWB.

Finally, although previous studies have verified that physical and mental health and social capital are relevant variables of SWB, the research on the complex causal relationship between the interdependence of the two and their joint effect on SWB is lacking. This study uses the fsQCA method to explore the influence of entrepreneurs’ physical and mental health at the individual level and social capital at the collective level on SWB from the perspective of configuration. It broadens the research field of SWB of entrepreneurs in terms of research methods. The results show that good physical fitness, optimistic and confident personality, trust in others, and friendly interpersonal interaction are interdependent and influence each other, which jointly drive entrepreneurs to obtain high well-being.

### 6.2. Practical Implications

This study’s results show that both physical and mental health and social capital have a positive impact on entrepreneurial SWB. Therefore, it is necessary to consider their physical and mental health and social capital accumulation to improve their entrepreneurial well-being and promote their entrepreneurial activities.

Entrepreneurship is a high-risk, high-pressure, and high-labor-intensity activity. An irregular lifestyle and long-term high-pressure state can cause physical and mental health issues among entrepreneurs. They are one of the special groups that deserve attention. At the personal level, entrepreneurs should try to avoid bad living habits and irregular eating habits, maintain long-term regular fitness exercises and good living habits, and have a fit body and abundant energy to manage high-pressure entrepreneurial activities. A good physical state and a good psychological state complement each other. Entrepreneurs should maintain positivity and pay attention to their physical and mental health, which is conducive to improving well-being and conducting entrepreneurial activities more smoothly. At the organizational level, companies should set up special health management departments. Professional medical staff should monitor the health of entrepreneurs and employees, establish basic health records, and conduct long-term health data analysis, which can play an important role in managing emergencies and preventing health problems. From a policy perspective, the government should not only issue entrepreneurial support policies and provide economic and platform support to entrepreneurs, but also pay attention to the physical and mental health of entrepreneurs, organize corresponding health education activities, and establish special psychological counseling centers for entrepreneurs. Such a holistic environment can help entrepreneurs relieve anxiety and psychological pressure and promote their well-being.

Entrepreneurs need sufficient capital, customers, human and material resources, and other social resources to meet the needs of the development of their enterprise. The accumulation of social capital will effectively improve entrepreneurial performance, which is conducive to the improvement of their economic and social benefits, to obtain high job satisfaction and well-being satisfaction. In the Chinese context, the acquisition of social capital is inseparable from the “collective concept” and “human society”, and the social capital of entrepreneurs is more derived from the family network environment of family members and relatives or the community network environment of neighbors and friends. Informal networks and social trust are crucial for entrepreneurs to obtain sufficient social capital. Therefore, when entrepreneurs conduct business activities, they should not only limit themselves to the enterprise network and formal organizational relationship, but should also focus on the perimeter of the close interaction and communication in their lives, moderate participation during family and community activities, establish intimate connections and sufficient trust with others, to prompt the production of mutually beneficial collective behaviors and contribute to the interconversion and effective utilization of resources. Establishing a community of interest with high trust intensity and strong emotional bonds is beneficial for the improvement of both the general economy and quality of life and obtaining higher life satisfaction and SWB.

### 6.3. Limitations and Future Research

This study has certain limitations. First, international differences among entrepreneurs were not considered. CGSS data only involve a sample of Chinese entrepreneurs, lacking global data for the corresponding analysis. Further, entrepreneurs may choose different antecedent conditions of SWB owing to the significant institutional, economic, and cultural differences among countries. Second, entrepreneurs are not classified into different stages of entrepreneurship. Entrepreneurs in different stages, such as start-up, survival, growth, and maturity, face different enterprise conditions and have different needs. For example, the initial and survival periods of entrepreneurs are more likely to require a lot of social capital to maintain business operations and reduce the risk of failure to be satisfied. However, mature entrepreneurs may no longer need the accumulation of too many connections and pay more attention to their physical condition and realization of self-worth; moreover, physical and mental health is the key to their well-being. Finally, there is no longitudinal research on entrepreneurial SWB in this study. SWB is a process of dynamic construction that fluctuates with the passage of time and changes according to the environment [106]. However, this study only uses CGSS2017 data to explore the SWB of entrepreneurs, which has certain research limitations.

Based on this, future research can consider the comparative study of entrepreneurs in different countries, which is conducive to a more detailed horizontal comparison of the factors influencing entrepreneurs’ SWB, to put forward specific suggestions and countermeasures for entrepreneurs in different countries. Further, heterogeneity research can also be conducted on entrepreneurs at different entrepreneurship stages to explore entrepreneurs’ different needs at different stages of the attainment of well-being. The most important point is that the dynamic change process of SWB should be considered, and the dynamic QCA method should be used to examine the key variables affecting entrepreneurs’ SWB in different periods, from a longitudinal perspective and using multiple sets of data. It is of great practical significance to construct a dynamic model of the influence of entrepreneurs’ SWB to help entrepreneurs obtain high SWB and effectively conduct entrepreneurial activities and create socioeconomic value.

## 7. Conclusions

This study aims to explore the combination of antecedent variables that influence entrepreneurs’ SWB. Based on CGSS2017 effective entrepreneurs’ sample data, the fsQCA method and positive psychology and self-efficacy theory were used to explore the causal relationship and configuration effect between entrepreneurs’ physical and mental health at the individual level and social capital at the collective level. The results show that physical health, optimism, self-efficacy, cognitive social capital, and structural social capital are not necessary for entrepreneurs to achieve high SWB, but they are sufficient conditions. The study sums up four different types of entrepreneurs to promote the SWB configuration path: optimistic efficiency-driven, trust efficiency-driven, strong psychology-driven, and weak relationship-driven. Our study confirms the significant correlation between physical and mental health, social capital, and SWB in the past and finds that there is a symbiotic relationship between the two dimensions, which jointly affect the SWB of entrepreneurs. Meanwhile, the rational analysis of the four configuration paths in the Chinese context can help Chinese entrepreneurs establish a weak relationship-oriented social network in the subsequent interpersonal interaction process.

## Figures and Tables

**Figure 1 ijerph-20-00417-f001:**
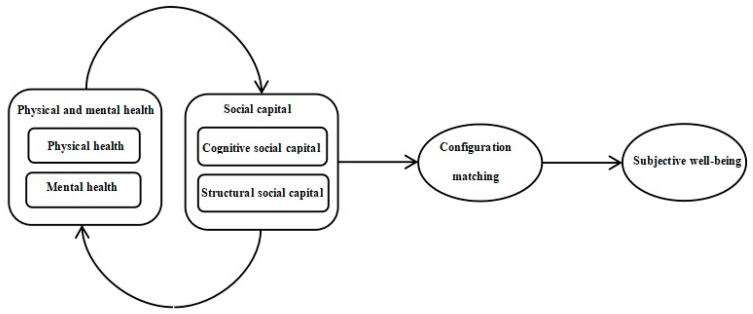
Theoretical model diagram.

**Table 1 ijerph-20-00417-t001:** Description of variables, metrics, and definitions.

Variable	Indicators	Definition
Subjective well-being (Result variable)	Subjective well-being (SWB)	Generally, how happy would you say your life is?
Physical and mental health (Antecedent condition)	Physical health (PH)	How do you feel about your current physical health?
	Mental health (SE)	If I find myself in a difficult situation, I can think of many ways to get out of it
		I am currently doing my best to pursue my goals
		There are many solutions to my problem
		I consider myself quite successful
		I can think of many ways to achieve my immediate goals
		Right now, I’m achieving the goals I set for myself
	Mental health (OM)	When things are uncertain, I usually hope for the best
		I am optimistic about my future
		On balance, I expect more good things to happen to me
		For me, if things can go wrong, they will
		I hardly ever expect things to go my way
		I rarely expect good things to happen to me
Social capital (Antecedent condition)	Cognitive social capital (ST)	Generally speaking, do you agree that most people in society can be trusted?
	Structural social capital (NI)	How often do you socialize with your neighbors?
	Structured social capital (FI)	How often do you engage in social entertainment with other friends?

**Table 2 ijerph-20-00417-t002:** Calibration of conditional and outcome variables.

Variable	Full Membership	Crossover Membership	Full Non-Membership
Subjective well-being	5.00	4.00	2.00
Physical health	5.00	4.00	2.00
Optimistic	4.50	3.67	2.83
Self-efficacy	6.83	5.17	3.00
Social trust	5.00	4.00	2.00
Neighbor interaction	7.00	5.00	1.00
Friends interaction	7.00	5.00	1.00

**Table 3 ijerph-20-00417-t003:** Necessity analysis of subjective well-being.

	Condition Variables	The Results of the Variable	
		High Subjective Well-Being	Non-High Subjective Well-Being
Physical and mental health	PH	0.808	0.714
	~PH	0.632	0.733
	OM	0.721	0.664
	~OM	0.644	0.707
	SE	0.742	0.647
	~SE	0.665	0.768
Social capital level	ST	0.653	0.596
	~ST	0.825	0.890
	NI	0.593	0.588
	~NI	0.735	0.745
	FI	0.698	0.668
	~FI	0.713	0.752

Note: PH = physical health, OM = optimism, SE = self-efficacy, ST = social trust, NI = neighbor interaction, FI = friend interaction, ledge breadth, “~” indicates the negation of the condition.

**Table 4 ijerph-20-00417-t004:** Configuration of high subjective well-being.

Condition Variables	The Results of the Variable			
	High Subjective Well-Being			
	1	2	3	4
PH	⚫	⚫	⚫	⚫
OM	●		●	⚫
SE	⚫	⚫	⚫	⚫
ST		⚫	⚫	◎
NI	◎	◎	◎	⚫
FI	◎	◎		⚫
consistency	0.925	0.974	0.971	0.946
coverage	0.398	0.356	0.341	0.320
Unique coverage	0.040	0.041	0.019	0.065
Consistency of solution	0.923			
The coverage of the solution	0.531			

Note: “●” indicates the presence of the condition, “◎” indicates the absence of the condition, and the space indicates that the condition may be present or absent; the core conditions are labeled as large “●” and large “◎”, and the edge conditions are labeled as small “●” and small “◎”. PH: physical health, OM: optimism, SE: self-efficacy, ST: social trust, NI: neighbor interaction, FI: friend interaction.

## Data Availability

The data that support the findings of this study are available from the corresponding author upon reasonable request.

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
