# Peer review of "What Are the Recipes of an Entrepreneur’s Subjective Well-Being? A Fuzzy-Set Approach for China"

_ijerph, 2022, doi:10.3390/ijerph20010417_

Round 1
Reviewer 1 Report
Thank you for submitting your article What are the recipes of entrepreneur's subjective well-being for review? A fuzzy-set approach for China.
The article has several weaknesses that should be improved.
The abstract should be prepared according to the requirements of the journal: (Background, Methods, Results, Conclusions).
Page 2 lines 53-54 - cited results appear, poorly communicated.
Correlation studies are described as longituidal, or experimental studies
The inrtroduction section is too chaotic for me, needs rewriting.
The operationalisation of the variables is questionable.
SWB is measured with one item. Other psychological variables such as self-efficacy are also measured with a random scale with no given psychometric properties.
The discussion includes the sentence ]. Based on the theory of positive psychology ... but no specific theory is given, and there is no bibliographic reference. Positive psychology has some assumptions, but it is not a theory in itself. It would be useful to indicate which theories are meant, which postulates of positive psychology the authors have in mind.
The article is chaotically prepared and needs more attention in the theoretical part and discussion. Showing why the use of single items is possible in this study. Pointing out their psychometric properties etc.
I am not familiar with the statistical method that was used in this study. The correctness of this part of the article should be assessed by someone who is familiar with the statistical methods used.
Author Response
Please see the attachment(Word)

Reviewer 2 Report
The article is interesting and presents a complex approach to the topic of well-being.
It is very well structured, with key ideas after each part.
The diagram of the theoretical model is relevant to the content of the article.
Line 241 - Is there no more recent data that can be interpreted? The data is already 5 years old.
Line 253 - Is there any other information about the entrepreneurs? Possibly their geographical distribution, etc.
Line 315 - I recommend the more detailed description of the FsQCA software.
The cultural implications and correlations described in subsection 6.1 are particularly interesting.
Author Response
Please see the attachment (Word)

Round 2
Reviewer 1 Report
The authors have made significant improvements to the article, and, in its current form, it can be published.